# *Fusarium* Mycotoxins Stability during the Malting and Brewing Processes

**DOI:** 10.3390/toxins11050257

**Published:** 2019-05-07

**Authors:** Karim C. Piacentini, Sylvie Běláková, Karolína Benešová, Marek Pernica, Geovana D. Savi, Liliana O. Rocha, Ivo Hartman, Josef Čáslavský, Benedito Corrêa

**Affiliations:** 1Biotechnology Department, University of Sao Paulo, Av. Professor Lineu Prestes, Sao Paulo 2415, Brazil; correabe@usp.br; 2Research Institute of Brewing and Malting, Malting Institute Brno, Mostecká 7, 614 00 Brno, Czech Republic; belakova@beerresearch.cz (S.B.); benesova@beerresearch.cz (K.B.); pernica@beerresearch.cz (M.P.); hartman@beerresearch.cz (I.H.); 3University of the Extreme Southern Santa Catarina, Av. Universitaria, Criciuma 1105, Santa Catarina, Brazil; geovanasavi@gmail.com; 4Department of Food Science, Food Engineering Faculty, University of Campinas, Av. Monteiro Lobato, 80., Campinas 13083-862, Brazil; lrocha@unicamp.br; 5Institute of Food Chemistry and Biotechnology, Faculty of Chemistry, Brno University of Technology, Purkyňova 118, 612 00 Brno, Czech Republic; caslavsky@fch.vut.cz

**Keywords:** mycotoxins, stability, malting, brewing, beer

## Abstract

Mycotoxins are widely studied by many research groups in all aspects, but the stability of these compounds needs further research for clarification. The objective of this study is to evaluate deoxynivalenol and zearalenone stability during all steps of the malting and brewing processes. The levels of these compounds decreased significantly during the production process (barley to beer). During the malting process, the DON levels decreased significantly in the steeping, germination, and malting steps (62%, 51.5%, and 68%, respectively). Considering ZEN, when the levels were compared between barley and the last step of the process, a significant decrease was observed. Most of the mycotoxins produced were transferred to the rootlets and spent grains, which is advantageous considering the final product. Furthermore, the mycotoxin dietary intake estimation was included in this study. The results proved that if the concentrations of target mycotoxins in raw material are under the limits established by the regulations, the levels decrease during the malting and brewing processes and make the beer secure for consumers. The quality of the five commodities involved in the beer process plays a decisive role in the creation of a safe final product.

## 1. Introduction

*Fusarium* contamination in cereals has increased in recent years, mainly in barley, wheat, corn, and oats [1,2,3,4,5]. This can be related to changes in climate condition, but is also due to problems during cultivation and harvest. Along with fungi contamination, temperature, and changes in forecast, the production of secondary metabolites occurs. 

Mycotoxins are being widely studied by many research groups in all aspects, such as the occurrence of these compounds in food, effects on animals and also on humans, prevention, and mitigation [2,6,7]. The stability of mycotoxins is also studied, but all aspects need further research for clarification and perhaps help in the control of contamination. 

In this context, the beer industry has several concerns about fungi and mycotoxin contamination due to the loss of raw material, economic impact, and, one of the most significant issues, loss of quality.

Two *Fusarium* toxins that are considered hazardous to the entire industry are deoxynivalenol (DON) and zearalenone (ZEN). These toxins are very common and can be found throughout the entire malting and brewing processes [8].

DON, also named vomitoxin, is a trichothecene B-type that elicits a complex spectrum of toxic effects. Chronic exposure to low doses of DON can lead to anorexia, impaired weight gain, and immunotoxicity. Acute exposure to high doses can cause diarrhea, vomiting, leukocytosis, circulatory shock, and, ultimately, death [9].

Trichothecenes are products of the sesquiterpenoid metabolism of some genera in the order Hypocreales [10] and, among them, deoxynivalenol (DON) is notorious for its worldwide occurrence. This mycotoxin is mainly produced by phylogenetic species within the *Fusarium graminearum* species complex, which includes the *F. graminearum sensu stricto* [11]. This species is encountered on barley, maize, and wheat causing Fusarium head blight, a disease that leads to yield losses in small grain cereals.

ZEN is another mycotoxin that should be taken into account due to the high contamination found in recent monitoring studies in grains [2,3,12]. It is treated as a non-steroidal estrogenic compound because its structure is similar to the hormone estrogen and it competes with 17β-estradiol for binding of the estrogen receptor, resulting in infertility and reproductive problems [13].

Regulations have been established worldwide to fix the maximum levels allowed of mycotoxins in grains. In Europe, the levels for DON and ZEN in barley are set at 1250 µg/kg and 100 µg/kg, respectively [14]. In Brazil, the regulations are being updated and in 2019 the DON levels for barley will be set at 1000 µg/kg for barley and 750 µg/kg for malt. Considering ZEN, the levels will be maintained at 100 µg/kg [15].

Some studies have shown a high contamination of these two mycotoxins in beer, but no studies have been developed to control or decrease the levels. The beer industry requires more information about the stability of these compounds during the entire process. 

For the reasons stated above, the objective of the current study is to evaluate the stability of DON and ZEN during all steps of the malting and brewing processes, considering that barley grains constantly pass through different conditions, which can increase or decrease the levels of these metabolites. To our knowledge, this will be the first study carried out considering mycotoxins in the entire beer chain.

## 2. Results

### 2.1. Method Validation

The methods for extraction of mycotoxin grains and liquid matrices during the malting and brewing process were validated according to the Commission Regulation [16] guidelines. To determine the recovery, repeatability, and selectivity/specificity, solid and liquid samples with non-detectable levels of mycotoxins were submitted to spiking experiments. 

The blank solid malt control sample was used (produced from the barley variety Kangoo) with no targeted mycotoxins detected. The spiking levels were chosen considering the contamination levels of examined samples. The blank liquid sample used was wort (from the above-mentioned malt, according to Section 4.3). The spiking levels of the liquid matrix were chosen considering the predicted contamination levels of examined samples of produced wort and beer. The standards were applied to the matrix in methanol:water (1:1 v:v). Average recovery was calculated using triplicate analyses conducted for each level. The experiments were repeated on three different days.

The limit of detection (LOD) was defined as the minimum concentration of an analyte in the spiked sample with a signal–noise ratio equal to 3 and limit of quantification (LOQ) with a signal–noise ratio equal to 10. Considering linearity, a seven-point calibration dependence was constructed with the following concentrations of the mycotoxin standard mixture (DON and ZEN): 5, 10, 20, 50, 100, 200, and 1000 ng/mL. The coefficients of determination (*r^2^*) for DON and ZEN were 0.9994 and 0.9992, respectively. The parameters achieved are shown in the Table 1.

### 2.2. DON and ZEN Stability during the Malting and Brewing Processes 

The stability of the mycotoxins was evaluated in the steps considered as the most important (*N* = 13) of the malting and brewing processes (Figure 1). First of all, as aforementioned, the barley used for the analysis was naturally contaminated and the levels among ten samples had an average of 3835 µg/kg and 1069 µg/kg for DON and ZEN, respectively (Table 2).

Considering DON in the malting process, water was analyzed first and an average of 599 µg/kg was found. In this experiment, the mass balance of mycotoxins DON and ZEN was always related to 1 kg of the original barley and the reported values in malt, wort, beer after boiling, and beer after fermentation were recalculated according to the weight balance, where barley is 100%.

Detected levels in byproducts (yeasts, spent grains, and rootlets) were reported in µg/kg and were not included in the mass balance.

Comparing DON levels found in barley with levels found on the first and third day of steeping and the third day of germination and malt, DON had a decrease of 66%, 62%, 51.5%, and 68%, respectively, and were significantly different (*** *p* < 0.001) (Figure 2). On the other hand, DON decreased significantly (38%) between the third day of germination and malt (* *p* < 0.05).

The ZEN levels during the malting process decreased significantly (79%) when barley was compared to the first day of steeping (* *p* < 0.05). Nevertheless, an increase could be seen between the first day of steeping and the third day of germination (173%), with a significant difference (** *p* < 0.01).

Further analyses were completed in brewing. Considering the entire process (barley to beer), DON showed a significant (*** *p* < 0.001) decrease of 71.6%. In the same context, when ZEN levels were compared, a significant decrease was also observed (** *p* < 0.01). 

It is necessary to highlight that all of the results for the liquid matrix were recalculated to 1 kg.

### 2.3. Mycotoxin Dietary Intake Estimation from Beer Consumption 

Using an average adult body weight of 60 kg and the mean mycotoxin data from this study, an estimation of dietary exposure to DON from the last step of the process while following the levels established by the regulation was carried out (Table 3).

It is important to mention that the level of the mycotoxin in the last step was 1089 µg, recalculated to 1 kg. Based on the literature [17], 1 kg of barley is able to produce 4 L of beer. 272 μg/L of DON in beer is equivalent to 1088 μg/kg DON in barley. This corresponds to a 71.6% reduction of DON content in the final product, whereas the average DON concentration in the original barley was 3835 µg/kg. Applying the same model to barley contaminated with 1250 µg/kg DON (the maximum allowable limit), the resulting DON concentration in beer will be 89 µg/L.

## 3. Discussion

As the literature shows, mycotoxins are highly stable compounds (resistant to high temperatures and extreme pH levels) [18]. Although the malting and brewing processes have maximum operation temperatures below the ones able to destroy the mycotoxins, they may influence mycotoxin concentration due to physical, chemical, and biochemical changes that take place [17,19]

In the current study, mycotoxin levels were determined in seven steps of the malting process. First, barley and water were analyzed and the results showed levels for DON in both matrices (barley average 3835 µg/kg, water average 599 µg/kg). The amount of DON found in the water can be explained by the soluble characteristic of this mycotoxin; for this reason, it was eluted from the matrix [20]. On the other hand, low levels for ZEN in water were found. 

Furthermore, when DON levels of barley were compared to the first and third day of steeping, a decrease in the levels was observed. These results could also be related to the loss of mycotoxins in the water.

The next step involved in malting is germination. In this study, the samples were taken from the first and third day of this process. The levels of DON decreased significantly between barley and the third day of germination. A possible reason for this reduction (DON glycosylation) is due to the increase in glucose content, which might activate the barley enzyme responsible for the respective reaction, transforming DON into DON-3-glucoside. Some studies have shown that 50% of DON is converted after a few days of germination [21]. To our knowledge, there is no study carried out considering ZEN glycosylation during germination in barley. However, the results of the current study could suggest this reaction (between barley and the first day of germination), transforming ZEN in α- and β-zearalenol and other masked toxins, even though the decrease in the levels is considered without significant difference [22]. 

Additionally, during germination, fungal biomass and mycotoxin production may grow, most likely as a result of cross-contamination from the residual steeping water or because of a latent barley grain infection, which may be activated with the increase in humidity during this process [23]. This can be seen in the current study when ZEN levels from the first day of steeping and the third day of germination are compared. 

The final procedure in malting is kilning, where the germination is interrupted and the green malt is prepared for storage and transportation. This step is characterized by several temperature scales (55–100 °C) and it is considered crucial to malt flavor and color formation [19]. In the current study, DON levels were shown to decrease significantly between the third day of germination and malt. This could be related to the high temperatures, but also due to the separation of the rootlets from the grain. 

Malt and rootlets were analyzed for both toxins and it is necessary to highlight the eminent amount of both toxins found in rootlets. For the beer industry, it was considered a good result, as most of the mycotoxins are released in this part of the grain. On the other hand, it should be a concern for agriculture, as this matrix is frequently used to feed animals [24,25,26].

After barley is malted, the grains are milled, and water is added to start the brewing process. In the mashing phase, the water is heated to about 62–71 °C and enzymes, such as α- and β-amylases, are activated to allow the conversion of starches into fermentable sugars [17]. Concerning the influence on mycotoxin levels at this step, there is a possible release of DON conjugated to protein structures and, as a result, there is an increase in total DON concentration [27]. This fact could be seen in the current study for both mycotoxins, not only for DON. Being more specific, DON and ZEN had an average malt level of 1211 µg/kg and 391 µg/kg, respectively. In mashing, if the results of wort and spent grains are added, the value is almost twice as high for DON and almost four times as high for ZEN (DON: 2173 µg/kg; ZEN: 1460 µg/kg). Also, the milling process could interfere in the levels due to the mycotoxin homogeneous spread into the entire malt batch and its later solubilization into mashing water [8].

Boiling is the next stage and it is characterized by the enzyme inactivation, hops addition, isomerization of hop alfa acids, evaporation of water and volatile compounds (dimethyl sulphides, undesirable in beer), protein precipitation, sterilization, Maillard reactions, and flavour modulation [17]. Regarding the impact of wort boiling on mycotoxin content, the temperature in this process, (above 100 °C) and the average time of boiling (1 h) could cause a decrease in mycotoxin concentration [8,28]. This experience can be seen in the results of the present study, even in low concentrations.

On the other hand, a few authors have shown that DON and ZEN are very stable compounds and the melting points in food processing are around 153 °C and 165 °C, respectively [18,29]. Another thing that should be mentioned in this stage is the addition of adjuncts, such as corn, wheat, and sorghum, which might represent an additional source of mycotoxins [1,30].

After the boiling process, the wort is cooled to 21 °C and the fermentation of wort is initiated by the yeast (*Saccharomyces* genus). In general, the fermentation temperature ranges from 2 °C to 30 °C, over 7–9 days [31,32]. DON was shown to be very stable in this process and the results did not show significant differences (after boiling: 1132 µg/kg; after fermentation: 1089 µg/kg = 3.8% of apparent reduction). In research performed by [33], similar results were found, where there was a reduction of only 11.6% of DON. 

The adsorption of mycotoxins by yeasts have been studied and this phenomenon occurs due to the binding of the toxins to ß-glucans from yeast to the cell wall [34]. In the current study, the yeasts were also analyzed after fermentation and DON was shown to be adsorbed. For ZEN, any amount of this mycotoxin found after boiling made this determination impossible in the present experiments. Nevertheless, in a study conducted by [33], ZEN was shown to have a very high ratio of adsorption (75.1%) on the beer fermentation residues.

Finally, considering the results of tolerable daily intake, the consumption of 62 L/year per capita [35] (0.17 L/day of beer) resulted in an exposure of 0.77 μg/kg/bw/day from the beer in this study and 0.25 μg/kg/body weight (bw)/day from beer using the levels established by the regulation for a 60 kg adult. Both these estimated daily intake (DI) results are still lower than the provisional maximum tolerable daily intake (PMTDI) of 1 μg/kg bw/day for DON stipulated by the Joint Expert Committee on Food Additives (JECFA) [36].

## 4. Material and Methods

### 4.1. Samples

A total of 10 barley samples (*Hordeum vulgare* L.) with a weight of 500 g each were used from a 2015 harvest and were supplied by the Brazilian agricultural research corporation (Embrapa). It is important to mention that the samples studied had natural mycotoxins levels acquired in the field without artificial interference.

The samples used in the current study were sown in May 2015 and harvested in December 2015. During this period, above-average rainfall with a mean precipitation of 95 mm was recorded [37]. Furthermore, the humidity was also considered significant, with an average value of 80% [38]. Another parameter included was the high temperature (average of 29 °C). These barley samples contained a concentration average of 3835 µg/kg for DON and 1070 µg/kg for ZEN, according to our previous study [2]; the environmental conditions presumably affected the mycotoxin production, thereby justifying the levels found.

The samples were used in the entire processes of malting and brewing (microscale) and aliquots were taken in the selected steps for mycotoxin analysis.

### 4.2. Chemicals and Reagents 

Deoxynivalenol and zearalenone standards were purchased from Sigma Aldrich (Vienna, Austria). The stock solutions standards were prepared in methanol:water (1:1) at a concentration of 1 μg/mL.

All reagents used in the following analyses were analytical and LC/MS-MS grade. For quality assurance in the mycotoxin analyses, wheat flour certified reference materials (CRM) (Trilogy, Washington) were used. The concentrations for DON and ZEN were 700 ± 100 μg/kg and 454.2 ± 37.6 μg/kg, respectively.

### 4.3. Malting and Brewing Process 

All of the malting and brewing trials were carried out in the Research Institute of Malting and Brewing, Brno, Czech Republic. The procedures used for malting were according to the [39] methodology and amendments were approved by the barley and malt European Brewery Convention [32].

The first step of malting was steeping, carried out as follows: On the 1st day of steeping, water was added to the grains for 5 h, followed by 19 h of air rest. On the 2nd day, grains were submitted to 4 h in the water, followed by 20 h of air rest. Finally, on the 3rd day of steeping, the grains were in the water for 20 min and then submitted to air rest for 23 h and 40 min. The germination step was performed over 3 days (72 h), with a subsequent kilning step. The grains in the steeping and germination processes were always maintained at a controlled temperature (14 °C) and the moisture content of the grains was measured in each step, being controlled to 45%. The total kilning time was 22 h, with a pre-kilning temperature of 55 °C for 12 h and a kilning temperature of 80 °C for 4 h.

Furthermore, the methodology used for wort production was according to [39] and the [32], with slight modifications. Fifty grams of milled malt was weighed into the mashing beaker and 200 mL of water (45 °C) was added. The beaker was placed into a programmed mashing bath with automatic stirrers. The temperature was maintained for 30 min under constant stirring (100 rpm). Following this period, the temperature of the mashing bath was increased 1 °C per minute for 25 min. Once the temperature reached 70 °C, 100 mL of water at 70 °C was added and maintained at 70 °C for 1 h. The beaker was then removed and cooled to room temperature. The contents of the beaker were adjusted to 450 g with water, stirred, and the contents filtered through filter paper. Wort and spent grains were used for further analyses.

The methods carried out for the final steps of the brewing process, such as boiling and fermentation, were developed by the Research Institute of Brewing and Malting and also according to available literature [31,32], in an attempt to simulate the real brewing process. In short, each sample with 200 mL of the wort was boiled with 8 g of traditional hops for one hour. It is necessary to mention that the hops were added at two different times, in the beginning (4 g) and after 30 min of boiling (4 g). The boiled wort was cooled to 21 °C for the yeast (W 34/70Starobrno Lager Yeast, *Saccharomyces pastorianus*) addition. The fermentation process was carried out over the course of 3 days at a controlled temperature (18 °C).

The sampling for mycotoxin analysis was performed in 8 steps of the malting process and 5 steps in the brewing process, as shown in Figure 1.

### 4.4. Extraction and Mycotoxin Cleanup 

To accomplish the procedure for extraction and cleanup of DON and ZEN analysis, immunoaffinity columns (DZT MS-Prep—R. Biopharm, Glasgow, Scotland) for both mycotoxins were used. This step was performed according to the DZT MS-Prep protocol with some modifications. In short, for grains, 10 g of milled barley, malt, rootlets, and spent grains were mixed with methanol 70% for 50 min, followed by 15 min of centrifugation at 4500 rpm. Then, 2 mL of the extract were added to 48 mL of PBS buffer (pH 7.4, adjusted with 2M NaOH) [40] and shacked manually. Twenty mL of the diluted extract were passed through the column at one drop per second. In addition, to wash the column, 20 mL of distilled water was added. The elution was performed with 2 mL of 100% methanol. For injection, the extract was dried and then dissolved with 1 mL of 50% methanol in water. For the yeast extract, the procedure described above was applied to 2 g per sample.

The DZT MS-Prep columns were also used for mycotoxin extraction from a liquid matrix. In summary, 2 mL of wort or beer was added to 18 mL of PBS buffer and then loaded to the column. For washing, 20 mL distilled water was used, followed by elution with 2 mL of 100% methanol. Finally, the extract was dried and dissolved in 1 mLof 50% methanol for injection.

### 4.5. LC/MS-MS Method for Analysis

To accomplish the identification and quantification of mycotoxins (DON and ZEN), LC/MS-MS system consisting of Finnigan Surveyor HPLC coupled to the ion trap LCQ Advantage mass spectrometer (Thermo-Fisher, USA) with atmospheric pressure chemical ionization (APCI) was used.

The chromatographic conditions were adopted according to the procedure developed by [41]. Chromatographic separation was performed with Synergi Hydro RP 80A column (3.0 × 150 mm, 4.0 μm particle size) equipped with a Security Guard™ cartridge C18 (4.0 × 3.0 mm, 4.0 μm) at 30 °C using gradient elution. The mobile phase was comprised of solvent A (water containing 10 mM ammonium acetate) and solvent B (methanol). The gradient program was applied at a flow rate of 0.5 mL/min under the following conditions: 0.1 min 90% A; 2 min 50% A; 10 min 20% A; 15 min 20% A; 16 min 90%; 25 min 90% A. The total analytical run time was 25 min for the two toxins.

The APCI interface was operated at negative polarity and the following ionization conditions were used: Capillary temperature, 160 °C; source heater temperature of 450 °C; nitrogen sheath gas flow of 35 L/min; nitrogen auxiliary gas flow of 10 L/min; source voltage of 6.0 kV; collision gas was helium. For selectivity, the mass spectrometer was operated in MRM mode and two transitions per analyte were monitored (Table 4).

The mass spectrometry conditions were optimized by re-tuning different analytes via direct infusion of each analyte individually. The cone voltages, collision energies, and product ions were optimized and carefully chosen.

### 4.6. Estimation of the Average Tolerable Daily Intake (TDI) and Maximum Tolerable Daily Intake 

The mycotoxin dietary intake estimation was calculated using the mean level of DON found in the end of the process (after fermentation/beer) which was divided by four, considering that 1 kg of barley is able to produce 4 L of beer [17]. The mean level obtained was multiplied by the daily consumption of beer according to [42] and divided by 60 kg (body weight). The TDI calculation was based on the tolerable intake 1 μg/kg body weight/day for DON [36].

### 4.7. Statistical Analysis

Results regarding DON and ZEN during the malting and brewing processes were reported as mean, maximum, and minimum using Microsoft Office Excel 2007. Also, analysis of variance (repeated measures ANOVA) using the Tukey test was conducted on the obtained data. The results are presented as mean ± standard deviation and values of *p* < 0.05 were considered statistically significant.

## 5. Conclusions

The malting and brewing processes may impact the stability of DON and ZEN. Both mycotoxins were shown to decrease in levels over the entire process (barley to beer), with significant difference. It is necessary to highlight that rootlets and spent grains were shown to discard most of the mycotoxins produced in the previous processes. However, it should be a concern for the agriculture, due to this matrix being destined for animal feeding.

The quality of the five commodities involved (barley, hops, water, yeast, and adjuncts) in the beer process plays a decisive role in the creation of a safe final product. Finally, according to mycotoxin dietary intake estimation, this study showed that if the raw material is under the limits established by the regulations, the levels can decrease during the process and make the beer safe for consumers.

## Figures and Tables

**Figure 1 toxins-11-00257-f001:**
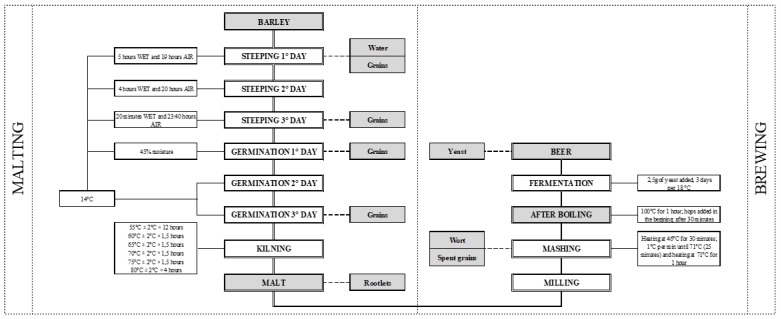
Map of malting and brewing process sampling (blocks in gray) carried out in the study.

**Figure 2 toxins-11-00257-f002:**
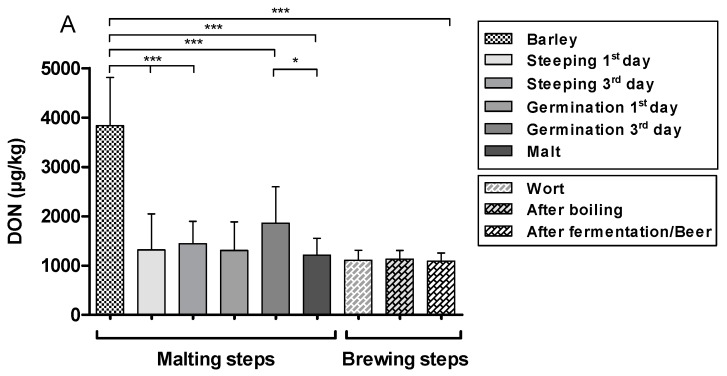
Steps of the malting and brewing processes showing (**A**) DON and (**B**) ZEN levels expressed as mean ± SD and different significance between the steps, according to ANOVA followed by Tukey test (* *p* < 0.05; ** *p* < 0.01; *** *p* < 0.001).

**Table 1 toxins-11-00257-t001:** Validation parameters of the method.

Analytes	Matrix	Spiking Level (ng/mL)	Recovery (%)	RSD ^c^ (%)	LOD *^d^	LOQ *^e^
DON ^a^	Grains (malt)	100	92.1	9.2	3.5	11.6
		500	92.9	2.9		
		1000	87.6	4.4		
	Liquid (wort)	50	90.6	1.4	0.5	1.6
		100	91.5	2.0		
		250	94.6	1.4		
ZEN ^b^	Grains (malt)	50	89.6	5.3	2.8	9.2
		250	99.8	2.9		
		500	105.7	0.4		
	Liquid (wort)	10	85.8	1.1	2.0	6.7
		50	89.2	0.9		
		100	97.9	0.8		

***** grain: µg/kg; liquid: µg/L; ^a^ DON: deoxynivalenol; ^b^ ZEN: zearalenone; ^c^ RSD: relative standard deviation; ^d^ LOD: limit of detection; ^e^ LOQ: limit of quantification.

**Table 2 toxins-11-00257-t002:** Average, minimum, and maximum values for all of the steps of the malting and brewing processes.

Process	Steps	DON (µg/kg)			ZEN (µg/kg)		
		Average	Maximum	Minimum	Average	Maximum	Minimum
Malting	Barley	3835	5204	2687	1070	3596	175
	Water 1st day	599	851	470	12	21	3
	Steeping 1st day	1319	2906	438	222	1045	53
	Steeping 3rd day	1442	2103	754	799	2602	15
	Germination 1st day	1308	2641	617	618	2385	57
	Germination 3rd day	1858	2957	822	1172	2854	140
	Malt	1211	1780	728	392	1386	95
	Rootlets	1797	2181	1431	1122	1735	206
Brewing	Wort	1105	1505	808	25	64	9
	Spent grains	1068	1742	590	1429	3188	200
	After boiling	1132	1483	840	<LOQ	<LOQ	<LOQ
	After fermentation/beer	1089	1414	863	<LOQ	<LOQ	<LOQ
	Yeast	166	241	89	<LOQ	<LOQ	<LOQ

**Table 3 toxins-11-00257-t003:** Mycotoxin dietary intake estimation from beer consumption.

	DON	
	Level Study	Level Regulation
Mean (µg/L)	272.17	88.75
Daily average exposure (µg/kg/bw)	0.77	0.25
Tolerable daily intake (µg/kg/bw)	1	1
% of tolerable daily intake	77	25

**Table 4 toxins-11-00257-t004:** Retention time and mass spectrometric parameters used in the analysis of the mycotoxins.

Mycotoxin	Retention Time (min.)	Precursor ion (m/z)	Product ion (m/z) *	Normalized CE (%)	Tube Lens
DON	6.3	355	295^C^	34	15
			265^Q^		
ZEN	16.2	317	273^C^	64	15
			299^Q^		

* C: Confirmation transition; Q: Quantification transition.

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
