# Peer review of "Fusarium Mycotoxins Stability during the Malting and Brewing Processes"

_toxins, 2019, doi:10.3390/toxins11050257_

Round 1
Reviewer 1 Report
The article got improved and many mistakes got corrected, but I still do not see any novelty in it. It is still unclear how was barley "contaminated with mycotoxins naturally ". Was the concentration of fungi so abundant that mycotoxins values were so high? Was it a rainy year during flowering? How did the infection occur? Where is the microbiological analysis? You are omitting to describe many important steps in materials and methods.
Author Response
1. The article got improved and many mistakes got corrected, but I still do not see any novelty in it. It is still unclear how was barley “contaminated with mycotoxins naturally“.
Response:
Thank you for your thoughts and observations. Although there is information regarding this subject, we supply more data which corroborate and validate the previous studies .
By naturally contaminated samples, the authors mean that the barley samples were not artificially infected with microscopic fungi either in the field or in the laboratory. This has now been updated and reflected in the manuscript (Ln 67-68).
The sample description was adjusted (Ln 69-74):
The samples used in the current study were sown in May 2015 and harvested in December 2015. During this period, above average rainfall with a mean precipitation of 95mm was recorded [16]. Furthermore, the humidity was also considered significant, with an average value of 80% [17]. Another parameter included were the high temperatures (average 29 °C). These barley samples contained a concentration average of 3835 µg / kg for DON and 1070 µg / kg for ZON, according to our previous study [2]; and the environmental conditions presumably affected the mycotoxin production, thereby justifying the levels found.
2. Was the concentration of fungi so abundant that mycotoxin values were so high?
Response:
The analysis of microscopic fungi in barley samples was not performed. We assume relatively high fungal contamination, relative to the detected mycotoxin levels.
3. Was it a rainy year during flowering?
Response:
The information about rainfall was added in the text. Ln70-71.
4. How did the infection occur?
Response:
Natural infection in the field due to climatic conditions.
5. Where is the microbiological analysis?
Response:
Microbiological analysis was not performed and was beyond the scope of this publication.
6. You are omitting to describe many important steps in materials and methods.
Response:
We added more information about samples and methodology.

Reviewer 2 Report
The authors failed to carry out a mass balance calculation and with the data presented, it is difficult to calculate the net effect of the different processing steps on the original amount of mycotoxins in the starting material (barley). The authors carried out ten independent trials from the starting material (barley grains) to the final product (beer). Then, authors should provide in the manuscript the following:
1/ In a typical malting process, the amount of malt (grams) obtained from 500 grams of barley
2/ In a typical brewing process, the amount of beer (grams or mL) obtained from 50 grams of milled malt. Also, the amount of wort (grams or mL) and spent grains (grams) is needed.
3/ According to 2.4., the amount of sample is known for grains (10 g) and liquid (2 mL for water and wort); please provide the sample amount for rootlets, beer and yeast.
Line 116-117. Better use ‘PBS buffer (pH 7.4, adjusted with 2M NaOH)’
Subheading 2.5 (new). In the first place, the estimation of intakes of mycotoxins is unreal because the original barley had a DON level above regulation and the beer manufactured by the authors under experimental conditions does not represent commercial beer in the market, which is the only one that can be taken into account to calculate intakes. Additionally, the redaction of 2.5 (and 3.3) is ambiguous and confusing. For the estimation of intakes from beer, I do not understand why DON level in beer is divided by four. The sentences in lines 203-207 generates many doubts ‘It is important to mention that the level of the mycotoxin in the last step was 1088 μg recalculated to 1 kg. Based on the literature [21] 1 kg of barley is able to produce 4 liters of beer. For this reason, the amount of 1088 μg/kg was divided per 4 liters, resulting in 272 μg/kg.’ What do you mean by this? What was the original concentration of DON in beer, 1088 µg/kg or 272 µg/kg? It is not clear if estimation of dietary intakes were calculated correctly.
TO AVOID DOUBTS AND MISINTERPRETATION OF THE RESULTS, IT WOULD HAVE BEEN NECESSARY TO CARRY OUT A MASS BALANCE CALCULATION.
2.6. The ANOVA type should be indicated (factorial design or repeated measures). A paired-test seems more appropriate than unpaired test; please indicate in the manuscript which one was used.
3.1. Method validation. In the present revised version, authors say that recovery experiments were done on barley samples (barley matrix) with non-detectable levels of mycotoxins that were spiked (‘grains’ in Table 2). However, in the reply to reviewers, it is said that recovery experiments were done ‘using the malt sample’, which one is correct? Additionally, in the revised manuscript there is no definition of what is the ‘liquid’ used for method performance (Table 2). In the reply to reviewers, it is said that the ‘liquid’ is wort, but why this important information is not included in the manuscript?
The analytical method is still questionable in terms of LOD and LOQ, and the reply to reviewers has not much improved in this sense. The presentation of this part is not fine: ‘The LOD and LOQ methods were performed (??) by fortifying blank samples with different concentration levels and the experiments were repeated on three different days’. What concentrations levels (please specify)? Please add to footnote to Table 3 the meaning of — (not detected? not analyzed?)
Author Response
Reviewer 2:
1) In a typical malting process, the amount of malt (grams) obtained from 500 g of barley.
Response:
From 500 g of barley approximately 420 g of malt is produced. Losses (malting yield, average root loss and breathing loss) need to be considered, about 16% in total, calculated for the original sample.
Recalculated for dry matter, the total losses are about 10 %. Note, that barley contains about 10 -12 % of water, malt about 4.5 – 5%.
Please see following reference: Hartman, Ivo: Quality of Malting Barley from Harvest 2013 in the Czech Republic, Kvasny Prum. 2015; 61(3): 69-74 | DOI: 10.18832/kp2015011.
2) In a typical brewing process, the amount of beer (grams or ml) obtained from 50 grams of milled malt. Also, the amount of wort (grams or ml) and spent grains (grams) is needed.
Response:
Methodology was included, Lns 101-108:
“Fifty grams of milled malt were weighed into the mashing beaker and 200 ml of water (45 ºC) was added. The beaker was placed into a programmed mashing bath with automatic stirrers. The temperature was maintained for 30 min under constant stirring (100 rpm). Following this period, the temperature of the mashing bath was increased 1 ºC per minute for 25 min. Once the temperature reached 70 ºC, 100 ml of water at 70 ºC was added and maintained at 70 ºC for a further 1 hour. The beaker was then removed and cooled to room temperature. The contents of the beaker was adjusted to 450 g with water, stirred, and the contents filtered through filter paper. Wort and spent grains were used for further analyses“
Further, for mashing, 50 grams of milled malt (each sample) was used. This produced about 100 grams of spent grains, which contained 70-80% of water.
3) According to 2.4, the amount of sample in known for grains (10g) and liquid (2 ml for water and wort), please provide the sample amount for rootlets, beer and yeasts.
Response:
The text was modified Ln 125.
4) Subheading 2.5 (new). In the first place, the estimation of interest of mycotoxins in unreal because the original barley had a DON level above regulation and the beer manufactured by the authors under experimental conditions does not represent the commercial beer in the market, which is the only one that can be taken into account to calculate intakes.
Response:
The main aim of our research was study of the stability of the targeted mycotoxins during the production of malt and beer in laboratory conditions. By calculating the dietary exposure, we wanted to indicate the potential risk of high beer consumption produced from contaminated raw materials.
5) Additionally, the redaction 2.5 (and 3.3) is ambiguous and confusing. For the estimation of intakes from beer, I do not understand why DON level in beer is divided by four. The sentences in lines 203-207 generates many doubts “It is important to mention that the level of the mycotoxin in the last step was 1088 ug/kg recalculated to 1 kg. Based on the literature [21] 1 kg of barley is able to produce 4 liters of beer. For this reason, the amount of 1088 µg/kg was divided per 4 liters, resulting in 272 µg/kg.” What do you mean by this?
Response:
From 1 kg of barley containing DON at a concentration of 3835 µg / kg, we can produce 4 l of beer with a DON concentration of 272 µg /1 l.
In this experiment, the mass balance of mycotoxins DON and ZON is always related to 1 kg of the original barley and the reported values in malt, wort, beer after boiling and beer after fermentation are recalculated according to the weight balance where barley is 100%. Detected levels in yeast and by-products (spent grains and rootlets) are reported in the ug/kg and they are not included in the mass balance.
This information was added into the manuscript.
6) What was the original concentration of DON in beer, 1088 or 272 ug/kg? It is not clear if estimation of dietary intakes were calculated correctly.
Response:
The concentration at the final stage was 272 ug/l, recalculated into the 1 kg of barley is 1089 ug/kg. Whereas the average DON concentration in the original barley was 3835 µg/kg, this corresponds to a 71.6% reduction of DON content in final product. This information was added into text.
7) TO AVOID DOUBTS AND MISINTERPRETATION OF THE RESULTS, IT WOULD HAVE BEEN NECESSARY TO CARRY OUT A MASS BALANCE CALCULATION.
Response:
For clarification, we used the calculation as described above (1 kg barley is 100 %)
8) 2.6 ANOVA
Response:
ANOVA was used as requested and the data updated and discussed in the text.
9) 3.1 Method validation: In the present revised version, authors say that recovery experiments were done on barley samples (barley matrix) with non-detectable limit of mycotoxins that were spiked (grains – table 2). However, in the reply to reviewers, it is said that recovery experiments were done “using the malt sample “, which one is correct? Additionally, in the revised manuscript, there is no definition of what is the “liquid“ used for method performance (Table 2). In the reply to reviewers, it is said that the liquid is wort, but why this important information is not included into manuscript?
Response:
Authors thank to reviewer and information were added into text, Lns 179-192.
10) The analytical method is still questionable in terms LOD and LOQ and the reply to reviewers has not much improved in this sense. The presentation of this part is not fine: “The LOD and LOQ methods were performed (??) by fortifying blank samples with different concentration levels and the experiments were repeated on three different days”
Response:
The text in the paragraph has been modified for better understanding, Lns 187-192.
11) What concentration levels (please specify)
Response:
Concentration levels have been summarized in the Table 2. We also calculated using the influence of the solid and liquid matrix, including dilution, weight and volume of the samples.
12) Please add to footnote to Table 3 the meaning of – (not detected, not analyzed?)
Response:
Table 3 was modified.

Reviewer 3 Report
This is a re-submission of a mansucript I previously reviewed. This version is improved on the points I stressed, except for the introduction that was not modified. This is a choice of the authors and their responsabiity.
In this version, mycotoxin dietary intake estimation from beer consumption had been added in Mat&Mat and in results. this improve the interest of the manuscript. unfortunaltely I did not find a discussion on this part. The author should add discussion on the 0.77 μg/kg/body weight in the samples you analysed compared to 1 μg/kg bw/day for DON stipulated by the FAO/WHO.
Finally dispite dilution, with naturally highly contaminated barley, there is a risk as the 0.23 μg/kg bw/day for reaching the limit could be in food from cereals...
Remarks for text editing :
line 48: with17β-estradiol ?
lines 184 and 196: Figure 2, not 1
Author Response
Reviewer 3:
1. This is a re-submission of a manuscript I previously reviewed. This version is improved on the points I stressed, except for the Introduction that was not modified. This is a choice of the authors and their responsibility.
Response:
The authors thank the reviewer for the favorable review and because two out of the three reviewers considered the introduction to be clear and compliant, the authors decided not to make any changes.
2. In this version, mycotoxin dietary intake estimation from beer consumption had been added into in Mat a Mat and in results. This improve the interest of the manuscript. Unfortunately I did not Find a discussion on this part. The author should add the discussion on the 0.77 ug/kg /body weight in the samples you analysed compared to 1 ug/kg bw/day for DON stipulated by the FAO/WHO. Finally dispite dilution, with naturely highly contaminated barley, there is a risk as the 0,23 ug/kg bw/day for reaching the limit could be in food from cereals….
Response:
Information added see above (Reviewer 2).
3. Remarks for editing:
Line 48: with 1,7 b-estradiol
Response:
Corrected
4. Line 184 and 196: Figure 2, not 1:
Response:
Corrected

Round 2
Reviewer 1 Report
You gave an explanation on climatic conditions during the year,that's OK. These values are skyhigh. The high temperatures are important for fungal growth, but they do not always equal high amounts of mycotoxins.
Determining both mycotoxins, DON and ZEA can be really tricky, and the polarities on LC-MS should be adjusted accordingly. I trust you had a great analytical expert for this.
Why did you measure mycotoxins in the 1st day's water? Final water is for sure abundant with DON.
Why do you mention the quality of 5 commodities, when you know only one aspect of one commodity (mycotoxins in barley) and you do not actually know the technicological quality of the used barley. BTW, such high contamination of mycotoxins purports extremely low quality of barley and non of the malting companies would admit such barley, naturally contaminated. It is extremely rare that such high mycotoxins concentrations even result in rootlets when DON is known as germination inhibitor.
Reviewer 2 Report
Thanks to the authors for the work done in the revision of their manuscript. I think that this review is much improved over the original version, and the comments and suggestions of the reviewers have been taken into account. In my opinion, the work is acceptable for publication in Toxins.
This manuscript is a resubmission of an earlier submission. The following is a list of the peer review reports and author responses from that submission.
Round 1
Reviewer 1 Report
The authors conducted a follow up on mycotoxins ZEA and DON during malting and brewing processes. Although done by LC-MS technique, this has been done before. So, the novelty and originality of this paper is alas very low. Recent publications dealt with this matter in a similar way and this represents just an information on mycotoxins in Brazilian barley-to-beer chain. You can check recent literature:
Habler, K.; Hofer, K.; Geissinger, C.; Schuler, J.; Huckelhoven, R.; Hess, M.; Gastl, M.; Rychlik, M. Fate of Fusarium toxins during the malting process. J. Agric. Food Chem. 2016, 64, 1377–1384.
Mastanjević, K.; Šarkanj, B.; Warth, B.; Krska, R.; Sulyok, M.; Mastanjević, K.; Šantek, B.; Krstanović, V. Fusarium culmorum multi-toxin screening in malting and brewing byproducts. LWT 2018, 98, 642–645, doi:10.1016/j.lwt.2018.09.047.
Mastanjević, Kristina; Šarkanj, Bojan; Krska, Rudolf; Sulyok, Michael; Warth, Benedikt; Mastanjević, Krešimir; Šantek, Božidar; Krstanović, Vinko. From malt to wheat beer: A comprehensive multi- toxin screening, transfer assessment and its influence on basic fermentation parameters. Food chemistry. 254 (2018) ; 115-121.
Mastanjević, Kristina; Šarkanj, Bojan; Šantek, Božidar; Mastanjević, Krešimir; Krstanović, Vinko. Fusarium culmorum mycotoxin transfer from wheat to malting and brewing products and by-products. World Mycotoxin Journal. (in press)
Results in these papers are remarkably relatable to your findings.
And there is also a review on mycotoxins in M/B by-products:
Mastanjević, K.; Lukinac, J.; Jukić, M.; Šarkanj, B.; Krstanović, V.; Mastanjević, K. Multi-(myco)toxins in Malting and Brewing By-Products. Toxins 2019, 11, 30.
... and many others...
Line 60: you say that the samples were naturally contaminated with mycotoxins... was the crop infected with Fusarium? Did you spike the barley with mycotoxins? How do you know that mycotoxins were present in the samples?
Line 96-97: why was the fermentation ended after only 3 days when it is usual to last for 7 days? Industrial conditions imply a 7-day fermentation, and the results for mycotoxins in spent yeast could be higher n case of 7-day fermentation.
Line 100-102 should be deleted.
Why have you presented results in tables and in graphical mode? Choose just one.
Line 214: you are mentioning activation of enzyme responsible "for the respective reaction, transforming DON into DON-3-Glucoside". Enzyme from what? Malt, Fusarium...? Please clarify.
Line 265: "The fermentation temperature ranges from 2 to 30 °C, during 7 – 9 2 days." Do you mean generally or in your research? Because in the materials and methods section you stated that the fermentation in your reaearch lasted for 3 days, if I got this right.
The results of your research are in accordance with other similar research but you failed to measure DON and ZEA metabolites which is very important since they transform into modified forms and that is especially problematic because they transform back into the basic form during digestion. English language should be checked by an English-speaking expert.
I hope this will be of help in case you decide to send it to another journal, but I have to suggest to the Editor to decline the publication in Toxins. Good luck!
Reviewer 2 Report
Abstract
The abstract should include global percentages indicating what fraction of the original DON and ZON present in barley is retained after malting and in the beer, for which a mass balance is needed. These numerical figures would be very indicative of the net effect of the malting and brewing process. Additionally, it is fine to indicate percent variations (not only concentrations in ppb) from different steps and sampling points.
Introduction
Lines 36-37: DON is primarily produced by two Fusarium species, which are usually denominated F. graminearum and F. culmorum. From what source have authors obtained the name Fusarium graminearum sensu strictu?
The paper of Habler et al. on Fate of Fusarium Toxins during Brewing (J Agric Food Chem, 2017, 65) is missing.
Material and methods
Line 58: The starting material for the malting and brewing assays was 10 samples of 500 g of naturally contaminated barley. Therefore, the brewing was done at microscale and this should be clearly mentioned somewhere in the manuscript.
Lines 67-69: What were the certified reference materials used (what cereal type)?
Subheading 2.3. The description of the malting and brewing process in the text is somewhat confusing, please try to improve it. Please clarify if the beer was 100% from barley or contained other mycotoxin-susceptible ingredients (adjuncts). Please specity the commercial name of yeast used for brewing (only Saccharomyces genus is not enough) as some yeast can degrade mycotoxins, or in turn, be negatively affected by mycotoxins.
The flow diagram in Figure 1 indicating in gray the 13 sampling points is fine, but a mass balance from the starting barley to the finished beer (including the fractions sampled and analyzed) is necessary for the purposes of comparison. Malting and brewing are complex processes with many steps in which the mycotoxins can be concentrated or diluted; sometimes there are large dilution factors to calculate the concentrations.
Lines 110-111: For IAC, could you check that the PBS buffer was made with 2M NaOH, pH 7.4?
Statistical: It is deduced from the text that there were 10 independent malting + brewing trials and therefore the average values in results section represent the mean of 10 results, is that right?
Results
Lines 148-149: For method performance, authors spiked/fortified blank samples with mycotoxins, but the nature of the blank samples is not explained (what grain and what liquid was used as blank samples?).
For DON the lowest spiking level in grains was 100 µg/kg, but the calculated LOD is almost 30 times lower (3.5 µg/kg). Similarly, lowest spiking level for ZON is 50 µg/kg, while the LOD is only 2.8 µg/kg. In my view, spiking levels for the calculation of recovery% and sensibility (LOD/LOQ) should include, at least, a level much closer to the target values. Additionally, calculated LOD/LOQ for both DON and ZEN in ‘grains’ and ‘liquid’ (Table 1) are below the lowest level (point) used in the calibration curves (5 µg/kg for both DON and ZON).
For the representability of results in Table 3, assuming that 10 independent trials were made, why not indicate the mean ± standard deviation for the ten results instead of the range (minimum and maximum)? Also, for clarity, round the results to zero decimals places. Again, for a proper interpretation of results, a mass balance would be very appropriate from 100% DON in the starting material to XX% DON left in the beer, and so for all the sampling steps.
For example, the concentration of DON in original barley was 3852 µg/kg and the concentration of DON in the first washing water is 599 µg/L, but what percentage of the original DON remained in the grains and what percentage is lost to the water?
Reviewer 3 Report
The manuscript reports data of an experiment where ten batches of barley with levels of contaminations in DON and ZON higher that the legal limits were followed all along malting and brewing processes for evaluating of the changes in concentrations of these mycotoxins. Considering mycotoxins in the whole beer chain is an originality of this work implemented with adapted analytical methods. However many works have been published in this topics and improvment in the manuscript would improve its interest.
Abstract and introduction should be improved for rendering the article more attractive.
In Abstract, instead of given selected data it could be presented evolutions of the content of mycotoxins: decreases due to solubilisation at steeping, increases at germination that could be due to new synthesis and release of masked forms, ….. And finally – 70 % in beer but risks for the by-products used for feeding animals. But actually all of this is known. You should highlight what is new in your work.
Introduction should be rewritten, with paragraphs longer than 1 sentence and a better introduction of DON and ZON including masked forms. Some of the problems associated with using FHB infected barley malt should also be presented in introduction, i.e. beer gushing.
Conclusion :
Lines 293-294: You should delete or edit this sentence. It is not a conclusion of your work. It should be true, but your study doesn't showed that. It was with samples having higher concentrations. Are there any linearity? You show 71 % reduction with samples all highest than the limits. And do you have any toxicological argument to state that 355 μg/kg (of grain used) is a con centration making the beer safe?.
According to EFSA, Tolerable Day Intake of 1μg/kg Body Weight for DON. Which give a concentration of =140 μg/L for a person of 70 kg BW drinking one 0.5 L bottle of beer per day. As you have measured DON in your final product, you should discuss this point.
Specific comments
In Abstract: Use . and not , as decimal marker
In introduction: concerning the highest contaminations levels in recent years, all the cited reference are from Brazil (1,2,3,9). Do you think that it is the same in other world areas?
Line 37: sensu stricto ?
Line 85: Which analyses? Wort production is not an anylsis.
Line 99: delete Results and the 3 following lines
Line 110: 2 ml of extract WERE added.
Line 187: ‘no levels’. ?? No ZON was detected or quantified.
Table 3: When you comment the data, could you add information on Max and Mini. Were the same samples at Max and mini all over the processes?